# Effects of Aerodynamic Parameters on Performance of Galloping Piezoelectric Energy Harvester Based on Cross-Sectional Shape Evolutionary Approach

**DOI:** 10.3390/mi16030254

**Published:** 2025-02-24

**Authors:** Xiaokang Yang, Bingke Xu, Zhendong Shang, Junying Tian, Haichao Cai, Xiangyi Hu

**Affiliations:** School of Mechatronics Engineering, Henan University of Science and Technology, Luoyang 471002, China; xbk1026@163.com (B.X.); jdszd@haust.edu.cn (Z.S.); tjy@haust.edu.cn (J.T.); chc1226@haust.edu.cn (H.C.); hxy230626@haust.edu.cn (X.H.)

**Keywords:** energy harvesting, piezoelectricity, galloping, aerodynamic parameter

## Abstract

This study explores the potential effects of the aerodynamic parameters on the performance of the galloping piezoelectric energy harvester. By considering the geometric configurations, a bluff body cross-sectional shape evolution approach is proposed using Boolean operations on the polygons and forty-eight different cross-sectional shapes with the protruding and depressed features are considered. Computational fluid dynamics is employed to perform a time-varying simulation of the aerodynamic characteristics. The effects of the aerodynamic parameters on performance are investigated computationally using a distributed parameter electromechanical coupling model. The critical wind speed, maximum output power, and the slope of the power versus wind speed curve are introduced as the performance evaluation parameters. The results show that the rear-side protruding feature and the top-side and bottom-side depressed feature have significant potential to enhance the performance. Furthermore, a symmetrical structure of the cross-sectional shape in the downstream direction should be prioritized over asymmetric designs.

## 1. Introduction

Micro-energy devices that convert ambient energy into electrical energy have been developed in recent years [1,2,3]. They can provide a continuous and stable energy supply for low-power electronic devices that are far from the power grid and can extend their service lives [4]. Piezoelectric energy harvesters [5,6], electromagnetic energy harvesters [7,8], triboelectric nanogenerators [9,10], and electrostatic energy generators [11,12] are the main types of micro-energy devices. Piezoelectric energy harvesters based on wind-induced vibration have been extensively researched due to the widespread availability of environmental wind energy, which is developed on a large scale [13]. Additionally, piezoelectric materials have high voltage output and are easily compatible with microelectromechanical systems [14]. Piezoelectric energy harvesters based on wind-induced vibration have vast potential for applications in large-span regional environmental parameter monitoring applications.

Structures in contact with airflow, whether natural or human-made, are inevitably subject to wind-induced vibration, which mainly includes galloping, vortex-induced vibration and buffeting, etc. [15] Galloping is a typical aerodynamic instability phenomenon with large-amplitude vibration and self-excitation properties [16]. This is beneficial when using the piezoelectric transducer because the output voltage is related to the amplitude of these oscillations [17]. The necessary condition for galloping is that the slope of the lift force coefficient is more negative than the drag force coefficient when the angle of attack of incoming flow is 0°. Structures with non-circular cross-sectional shapes are likely to experience galloping. Ice-covered wires, high-rise buildings, lamp poles, chimneys, and towers are typical structural types. Under the action of airflow, the total damping of the elastic system consists of linear damping and nonlinear damping. The former is composed of structural damping and linear aerodynamic negative damping, and the latter is nonlinear aerodynamic negative damping. When the wind speed exceeds a certain threshold (critical wind speed), the linear aerodynamic negative damping causes the total damping of the system to be negative, and the bluff body produces a single-degree-of-freedom divergent vibration with crosswind bending, while the nonlinear aerodynamic negative damping causes it to produce limited motion.

The common structural characteristics of the galloping-based piezoelectric energy harvester (GPEH) are a bluff body and a piezoelectric composite beam [13]. The bluff body drives or its galloping wake induces the vibration of the piezoelectric composite beam, and an electrical output is obtained on the load. The purpose of using mathematical models and experiments to analyze the effects of parameters on performance is to obtain the best structural design. The parameters of the GPEH mainly include aerodynamic, structural, motion, and electrical parameters. The galloping aerodynamic coefficient, which is obtained based on experiments or simulation processes, is mainly determined by factors such as cross-sectional shape, turbulence intensity, and Reynolds number. It is usually expressed as a function of the aerodynamic lift and drag coefficients and the angle of attack of incoming flow. The structural, motion, and electrical parameters mainly include the dimensional characteristics of the bluff body and piezoelectric composite beam, equivalent mass, natural frequency, mechanical and electrical damping ratio, electromechanical coupling coefficient, capacitance, and optimized load, etc. The effects of the structural, motion, and electrical parameters on performance can be analyzed adequately using the model; however, this is not applicable to the aerodynamic parameters, which are mainly determined by the cross-sectional shape of the GPEH’s bluff body. Manually changing the aerodynamic parameters will result in an unknown cross-sectional shape.

A reliable and accurate mathematical model is a prerequisite for the development of an optimal GPEH. The three commonly used types of mathematical models include the single-degree-of-freedom lumped parameter model, the approximated distributed parameter model based on Rayleigh–Ritz discretization, and the distributed parameter model based on Euler–Bernoulli beam theory [18]. The Gauss law and quasi-steady hypothesis are used to relate the mechanical and electrical variables and to evaluate the galloping force, respectively. The coupled nonlinear electroaeroelastic distributed-parameter model based on Euler–Bernoulli beam theory was developed by Abdelkefi et al. [19] in 2013, and it was validated using the previous experimental results of Sirohi and Mahadik [20]. The effects of the load resistance and wind speed on the performance of the GPEH have been studied, and the results showed that the onset(critical) speed is strongly affected by the electrical load resistance. Zhao et al. [21] developed a comparison study on the performance of the modeling methods, including the single-degree-of-freedom lumped parameter model and the single mode and multimode Euler–Bernoulli distributed parameter models. The influence of the load resistance, wind exposure area, mass of the bluff body, and length of the piezoelectric sheets on the cut-in(critical) wind speed, as well as the output power level, were fully investigated. It was demonstrated that all the considered models could accurately predict the performance of the harvester.

The research on the effects of aerodynamic characteristics on the performance of GPEHs based on the different cross-sectional shapes is well developed and includes studies of shapes that are square [22], rectangular [23], triangular [24,25], D-shaped [22], elliptical [26,27], V-shaped [28], W-shaped [29], Y-shaped [30], fork-shaped [31], sinusoidal wavy protuberances on square [32], circular with different rod-shaped attachments [33], and rectangular, triangular, and elliptical metasurface protrusions [34]. Their results confirmed the superiority of the square cross-sectional shape compared to the rectangular, triangular, and D-shaped; sinusoidal wavy protuberances on a square bluff body could be used as a passive control to tune the critical galloping velocity; the rear protrusions of rectangular, triangular, and elliptical metasurfaces can enhance the galloping response; and the power generated by the square bluff body with a specific ratio of V-shaped groove depths is much higher than that of the square bluff body at the same wind speed. A common characteristic of these studies is that the irregular changes in cross-sectional shapes were used; consequently, the conclusions were non-universal, which makes it difficult to design a structure guided by specific performance requirements.

In this study, an evolutionary approach to the bluff body cross-sectional shape of the GPEH is proposed, with the aim of systematically investigating the potential effects of aerodynamic parameters on performance; furthermore, the protruding and depressed features around the square bluff body are considered separately. The model, which was proposed in our previous work, is validated in Section 2 by the experimental results of a prototype with a square cross-sectional shape. In Section 3, computational fluid dynamics is employed to perform the time-varying simulation of aerodynamic characteristics for forty-eight bluff bodies with different cross-sectional shapes. In Section 4, the effects of aerodynamic parameters on performance are analyzed using the model. A summary and the conclusions are presented in Section 5.

## 2. Mathematical Formulation of GPEH

The GPEH comprises a thin piezoelectric cantilever beam rigidly attached at the free end to a bluff body with different cross-sectional shapes [17]. A distributed parameter electromechanical coupling model of a piezoelectric energy harvester based on the interaction between vortex-induced vibration and galloping has been proposed and verified [35]. The distributed parameter model of the GPEH can be obtained under non short circuit conditions and by ignoring the aerodynamic force of vortex-induced vibration [36], and the first-order model can be expressed as follows:(1)$$\ddot{Y}\left(t\right)+2\omega \xi \dot{Y}\left(t\right)+\omega^{2}Y\left(t\right)+\Theta V\left(t\right)=\frac{1}{2}\rho u^{2}LD\left[\sum_{i=1}^{N}A_{i}{\left(\frac{\dot{Y}\left(t\right)}{u}\right)}^{i}\right],$$(2)$$\frac{V\left(t\right)}{R}+C_{p}\dot{V}\left(t\right)=\Theta \dot{Y}\left(t\right),$$
where ω, ξ, and Θ represent the undamped natural frequency, mechanical damping ratio, and piezoelectric coupling term, respectively; $A_{i}$ represents the polynomial coefficients of the galloping aerodynamic force coefficient, L and D are the height and length of the bluff body; V represents the voltage across the load resistor R; and $C_{p}$, u, and ρ represent the total capacitance and wind velocity, respectively.

The time marching using ODE45 is used to obtain the numerical solutions of the displacement or electrical responses, and the flow chart of the simulation process is shown in Figure 1. Prior to this, the accuracy of the initial and convergence conditions must be validated. A slight displacement of the beam should be established as an initial condition for galloping. The convergence condition necessitates that the amplitude error of the aerodynamic coefficient or displacement response of the bluff body between two adjacent time steps must be less than 10^−5^, or until the calculation time reaches its maximum value. An updated initial amplitude should be considered, which means that the initial amplitude of each velocity calculation point is taken to be equal to the stable amplitude of the previous one, because the hysteresis characteristics of the galloping would not be predicted by considering a fixed initial amplitude.

The distributed parameter model was first validated using the experimental results of the prototype. The prototype was fabricated and tested in a small tunnel, as shown in Figure 2a. The bluff body was constructed from balsa wood with the sides of 4 × 4 × 10 mm and a mass of 0.0291 g. The cantilever was assembled using a 20 μm thick ultraviolet-curing process to attach 200 μm thick polyvinylidene fluoride (PVDF) films to 300 μm thick polyethylene terephthalate (PET) films. According to the model, the numerical first natural frequency, capacitance, and optical load are 97.29 Hz, 132.8 pF, and 12.32 MΩ, respectively. The experimental values are 98.8 Hz, 130.5 pF, and 13 MΩ, as determined using the amplitude–frequency characteristic, impedance analyzer (Microtest-6378, Taipei, China), and load characteristic, respectively, as shown in Figure 2b,c. The numerical results of those parameters agree well with the experimental ones. The numerical equivalent mass and piezoelectric coupling term are 0.03082 g and 1.935 × 10^−6^ N/V, respectively. The mechanical damping ratio was measured using the logarithmic decrement method [20,37], with values of 1.60–1.87%. The damping is amplitude-dependent [22] and is positively correlated with the amplitude of our experimental results. A variable damping ratio was applied to the calculation process, and a linear fit was used to express the relationship between damping and wind speed, as follows: ξ=0.00123u+0.0044. The experimental results and numerical results with fixed (1.74%) and variable damping ratios are shown in Figure 3a. The critical wind speed of the experiment is 9.2–9.4 m/s. Once the wind speed exceeds the critical wind speed, the load obtains a larger electrical response. For example, when the wind speeds are 10, 12, and 13.2 m/s, the experimental output powers are 23.1, 26.3, and 27.4 μW, respectively. In the working wind speed range, the experiment power versus wind speed curve is divided into linear and nonlinear regions, and the wind speed ranges are 9.4–11.2 m/s and 11.2–13.2 m/s, respectively. The nonlinear response region may be caused by material and geometric nonlinearity at the relatively high wind speeds. According to previous studies, the geometric nonlinearity of flexible cantilever beam systems under large displacements should be considered for its impact on bluff body motion [38]. This nonlinear strain–deflection relationship will lead to an amplitude reduction trend. Additionally, the piezoelectric material nonlinearity can induce a softening behavior in the system [39]. Therefore, higher-order elastic and electroelastic tensor components in the piezoelectric constitutive equations should be considered during the modeling process. The current model exhibits discrepancies in predicting electrical responses under high wind speeds compared to experimental results, primarily due to the exclusion of nonlinear factors in the modeling framework. With the variable damping ratio, the model accurately predicts the critical wind speed and the electrical response level of the linear region. Since the model does not consider nonlinear factors, the numerical output powers under high wind speed are higher than the experimental values. For example, when the wind speed is 13.2 m/s, the numerical and experimental output powers are 33.7 (fixed damping ratio), 31.1 (variable damping ratio), and 27.4 μW, respectively, with relative errors of 23.0% and 12.4%. In addition, the prototype has a slight electrical output below the critical wind speed, which may be caused by the non-strict symmetry structure of the prototype.

In 2013, Yang et al. [22] proposed a prototype of the GPEH with a square cross-sectional shape, and a single-degree-of-freedom lumped parameter model was used for the prediction of the electrical response after the identification of the prototype’s parameters. The equivalent mass, natural frequency, damping ratio, piezoelectric coupling term, capacitance, and optimized load are 30.168 g, 6.84 Hz (short circuit) and 6.8 Hz (open circuit), 1.48% (0.5% was used), 0.000373 N/V, 180 nF, and 105 kΩ, respectively. Figure 3b shows a comparison of the predicted results from the current model with the numerical and experimental and simulation data in Yang’s study, and the results of the two models are almost the same in terms of critical wind speed and output voltage.

## 3. Cross-Sectional Shape Evolutionary Approach and Simulation Results

### 3.1. Evolutionary Approach

GPEHs with different bluff body cross-sectional shapes have been analyzed theoretically and experimentally. Due to its accessibility, the GPEH with a square cross-sectional shape has traditionally been used as the subject of performance comparison. In fact, “a complex cross-sectional shape” is equivalent to a simple geometric pattern, such as a square, triangle, or circle, with variable external contour features. The variable external contour features can also be re-textured by performing Boolean operations on multiple simple geometric patterns. This means that they can be obtained through Boolean operations between multiple simple cross-sectional shapes; for example, a D-shape can be computed using Boolean union regions between a square and a circle. Furthermore, as the size of the outer contour feature gradually decreases, the complex cross-sectional shape will evolve towards a simple one. The evolutionary approach detailing the cross-sectional shapes discussed in this study is shown in Figure 4 and follows the given procedure.

(i)Primary and secondary cross-sectional shapes, e.g., a square and a circle, are selected.(ii)The Boolean difference operation is performed on multiple primary cross-sectional shapes to obtain the auxiliary cross-section shape, e.g., a rectangle, or by selecting the primary cross-sectional shape directly as the auxiliary cross-section shape, e.g., a square.(iii)The Boolean union or difference operation is performed on the auxiliary cross-sectional shape and multiple secondary cross-sectional shapes to obtain the final cross-sectional shape, e.g., a D-shape, and it evolves towards the primary shape as the number of secondary shapes approaches infinity.(iv)The final shape is named “$ROUGH_{F/R/FR/T/B/TB/FRTB}^{i-P/D}$”, with the superscript and subscript denoting the number and direction of the protruding (P) or depressed (D) features, respectively. For example, $ROUGH_{FRTB}^{4-P}$ means a square cross-sectional shape with the protruding features of four semicircles on the front (F), rear (R), top (T), and bottom (B) sides.

In fact, the evolutionary approach we proposed represents a general method for texturing the bluff body cross-sectional shape of the GPEH and also applies to scenarios such as those in which a square or circle with an equilateral triangle or arched features is used. This approach embodies the logic of the way in which a complex cross-sectional feature evolves towards the simplest and most readily obtainable shape according to a certain pattern. This makes it easier to capture the effect of the cross-sectional shape with the continuous feature changes on aerodynamic characteristics. Consequently, it facilitates the obtainment of the effects of aerodynamic parameters on the behavior, critical wind speed, and electrical response of GPEHs, and thus benefits their structural design. The forty-eight cross-sectional shapes discussed in this study are shown in Figure 5.

### 3.2. Aerodynamic Dynamic Modeling and Validation

To obtain the aerodynamic force on the bluff body, the external flow field is simulated by solving the continuity and the Navier–Stokes equations of incompressible fluid, with the assumption that the external flow field is 2D and unsteady [25,34]. As shown in Figure 6a, the computational domain is rectangular with a size of 70D × 20D, where D is the characteristic dimension of the cross-section of the bluff body. The distances from the center point of the bluff body to the inlet and top wall are 20D and 20D; these distances ensure adequate fluid development and a smaller blocking rate. The settings of the boundary conditions include the velocity inlet, pressure outlet, and symmetric top and bottom walls. A circular mesh refinement area around the bluff body is set, with a diameter of 5D. The height of the first layer of the mesh depends on y+, which needs to be close to 30 for the validity of the standard k−ε model in the near-wall region. The characteristic length of the bluff body is 0.06 m and the Reynolds number is about 12,322. The time step in the domain-independence study is finally set to 0.0005 s, meeting the requirement of the maximum Courant number to be less than unity.

The reference values are used in the computation of the derived physical quantities and non-dimensional coefficients for postprocessing, such as the lift and drag force coefficients. The accuracy of the calculation results is related to the reference value setting. The reference values, which are defined in ANSYS Fluent^®^ 2023 R1, mainly include the area, depth, length, and velocity. To aid understanding of the definition of the reference value, the height variable is introduced in this study. The height and length are the projection dimension of the bluff body in the cross-flow and downstream directions, as shown in Figure 6a, respectively. The depth is the thickness of the bluff body, and its value also needs to be set in 2D simulations. Finally, the area is the projection of the bluff body in the cross-flow direction, which is equal to the product of the height and depth. According to the above definition, the three-dimensional characteristics of the bluff body can be uniquely determined. Three different grid densities are considered to ensure the grid independence. The comparisons of results of the aerodynamic lift and drag coefficients, *C_L_* and *C_D_*, in the time domain are given in Figure 6b,c, when the angle of attack, wind speed, Reynolds number, and turbulence intensity are 0°, 15.3 m/s, 20,000, and 12.5%, respectively. The root mean square (RMS) values of *C_L_* and *C_D_*, given in Table 1, are consistent with the experimental results in Laneville’s study [40], and the medium grid number has been selected.

### 3.3. Simulation Results of Aerodynamic Characteristics of GPEH

According to the quasi-steady theory proposed by Parkinson [16], the lift and drag coefficients, *C_L_* and *C_D_*, of galloping, in the course of oscillation, are the same at each attack angle *α* as the values measured at the corresponding steady attack angle *α*, and the transverse force coefficient *C_Fy_* is defined as follows:(3)$$C_{Fy}=-C_{L}cos\alpha -C_{D}sin\alpha =-\left(C_{L}+C_{D}tan\alpha \right)sec\alpha ,$$

In terms of the Den Hartog criterion, a necessary condition for the occurrence of the galloping instability is that the slope of the *C_Fy_* versus *α* curve at *α* = 0° is positive, and can be expressed as follows:(4)$${\left.\frac{\partial C_{Fy}}{\partial \alpha }\right|}_{\alpha =0^{{}^{\circ}}}=-{\left.\left(\frac{\partial C_{L}}{\partial \alpha }+C_{D}\right)\right|}_{\alpha =0^{{}^{\circ}}}>0,$$

The numerical results for *C_L_*, *C_D_*, and *C_Fy_* of the forty-eight different cross-sectional shapes are shown in Figure 7 and Figure 8, in which the protruding and depressed feature cases are considered. Due to the asymmetric structure in the cross-flow direction, both the positive and negative angles of attack are considered, with a range of −30–30°. The aerodynamic characteristic analysis includes the protruding features on the front and rear sides and the depressed features on the top and bottom sides of the cross-sectional shape.

#### 3.3.1. Protruding and Depressed Features on the Front and Rear Sides

For the protruding feature, as shown in Figure 7a–c, the variation trend of the drag coefficient *C_D_* for the cross-sectional shapes with the front protruding feature is consistent with that of the square one. The values of *C_D_* at all the considered attack angles *α* are smaller than those of the square, and *C_D_* increases with the number of semicircular features at the same attack angle. This trend is easy to foresee, as the smooth front-side protruding feature causes its *C_D_* to decrease rapidly. Meanwhile, the corresponding values of *C_L_* are all larger than those of the square cross-sectional shape and are even positive. This causes the slope of the aerodynamic coefficient *C_Fy_* versus *α* to be negative at *α* = 0°, and it also causes *C_Fy_* to gradually decreases with the increase in *α*. Therefore, as shown in Figure 7c, no galloping behavior occurs for cases of $ROUGH_{F}^{1-P}$, $ROUGH_{F}^{2-P}$, $ROUGH_{F}^{4-P}$, $ROUGH_{FR}^{1-P}$, $ROUGH_{FR}^{2-P}$, and $ROUGH_{FR}^{4-P}$, and the GPEH with those cross-sectional shapes will not produce electrical output. It is interesting to note that for the rear-side protruding feature, *C_L_* and *C_D_* have more lower values at the larger attack angles; this results in the peak value and the attack angle corresponding to *C_Fy_* = 0 being clearly larger than the square shape. According to the previous research conducted by Xing et al. [34] and Hu et al. [33], the larger peak value reflects a greater galloping aerodynamic force and displacement response, which is beneficial to the performance improvement of the GPEH.

The depressed feature, as shown in Figure 7d–f, does not have a significant effect on *C_L_* and *C_D_*, and these minor effects are ultimately not reflected in *C_Fy_*. It is worth noting that compared to the square shape, the shape with the front-side depressed features has a larger *C_D_*. For example, when *α* = 0°, the *C_D_* values of the shapes of the depressed feature with the semicircular numbers 1, 2, 4, and 8 and the square shape are 1.75, 1.87, 1.70, 1.58, and 1.58, respectively. Within the considered range of *α*, *C_D_* is between 1.42 and 2.12, and its contribution to *C_F_*_y_ is small. However, the slopes of *C_Fy_* versus *α* at *α* = 0° are all positive, indicating that all these shapes will experience galloping.

**Figure 7 micromachines-16-00254-f007:**
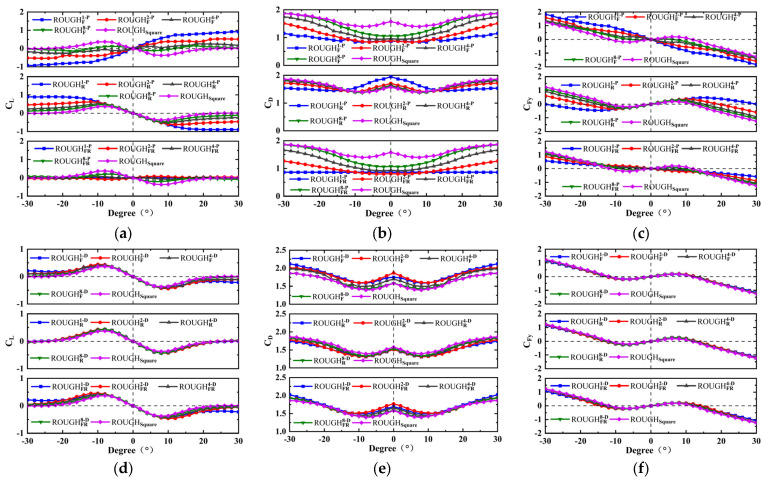
Numerical coefficients versus the angle of attack for protruding and depressed features on the front and/or rear sides: (**a**) *C_L_*, (**b**) *C_D_*, and (**c**) *C_Fy_* for the protruding feature; (**d**) *C_L_*, (**e**) *C_D_*, and (**f**) *C_Fy_* for the depressed feature.

#### 3.3.2. Protruding and Depressed Features on the Top and Bottom Sides

Due to the fact that the shapes with the top or bottom side features are only asymmetric in the cross-flow direction, the coefficient curves of *C_L_*, *C_D_*, and *C_Fy_* versus *α* of these shapes are not symmetric, with the origin point of coordinate axis in the range −30° < *α* < 30°. Therefore, for those cases, the effects of the protruding and depressed features on *C_L_*, *C_D_*, and *C_Fy_* can only be discussed in the range 0° < *α* < 30°. For the protruding feature, as shown in Figure 8a–c, the *C_L_* and *C_D_* values are higher and lower, respectively, than the square shape in most cases and are mainly caused by the smooth top or bottom protruding feature cases. This results in the slope of *C_Fy_* versus *α* at *α* = 0° and the peak value of the *C_Fy_* versus *α* curve are slightly lower than those of the square shape. The *α* corresponding to *C_Fy_* = 0 of the square shape is the same as that of the shapes with only top-side or bottom-side features, but larger than that of the shapes with both top-side and bottom-side protruding features. This indicates that the bluff body has smaller galloping oscillations, which is not conducive to the performance improvement of the GPEH with the above cross-sectional shapes. No galloping behavior occurs for cases of $ROUGH_{B}^{1-P}$, $ROUGH_{TB}^{1-P}$, $ROUGH_{TB}^{2-P}$, $ROUGH_{TB}^{4-P}$, and $ROUGH_{TB}^{8-P}$.

An important difference is shown by the depressed feature, as shown in Figure 8d–f. The values of *C_L_* and *C_D_* with the top-side depressed feature are almost the same in 0° < *α* 30°. This is because when *α* is greater than 0°, the top depressed feature is located on the rear side of the shape and has almost no effect on *C_L_* and *C_D_*. The bottom depressed features have limited effects on *C_L_* and *C_D_* at a smaller *α* (such as 0–10°), but their effects in a larger *α* (such as 10–30°) are obvious. According to the results, it can be seen that the degree of influence is positively correlated with the depth of the depressed feature. For example, as shown in Figure 9f, the peak values of *C_Fy_* and the *α* corresponding to *C_Fy_* = 0 of the depressed feature with the semicircular numbers 1, 2, 4, and 8 and the square shape are 0.88 (10°), 0.41 (8°), 0.27 (6°), 0.24 (6°), 0.20 (6°), and 18–20°, 12–14°, 10–12°, 10–12°, 10–12°, respectively. Therefore, several conclusions can be drawn, as follows:(i)A symmetrical shape with the depressed feature on the top and bottom sides is necessary.(ii)The slopes of the aerodynamic coefficient *C_Fy_* versus attack angle *α* curve at *α* = 0° of the depressed feature shapes are almost the same as that of the square, which leads to an almost consistent critical wind speed.(iii)The degree of influence is positively correlated with the depth of the depressed feature; therefore, the case of $ROUGH_{UD}^{1-CV}$ has a larger galloping displacement response, and the GPEH with this cross-sectional shape has a much higher electrical output.

**Figure 8 micromachines-16-00254-f008:**
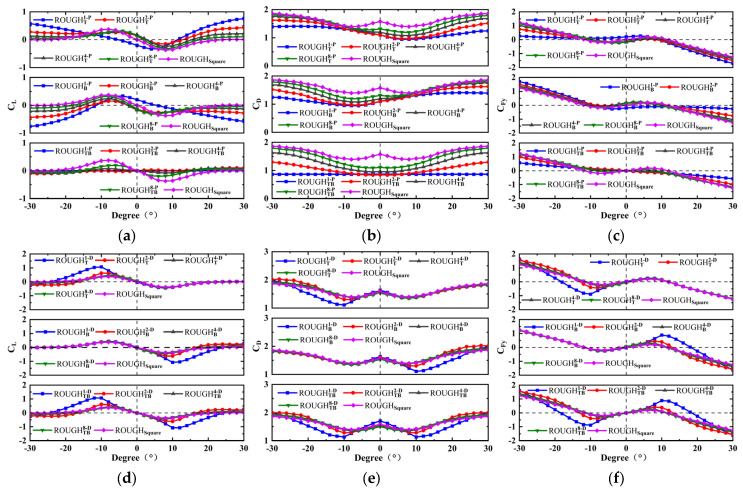
Numerical coefficients versus the angle of attack for protruding and depressed features on the top and/or bottom sides: (**a**) *C_L_*, (**b**) *C_D_*, and (**c**) *C_Fy_* for the protruding feature; (**d**) *C_L_*, (**e**) *C_D_*, and (**f**) *C_Fy_* for the depressed feature.

**Figure 9 micromachines-16-00254-f009:**
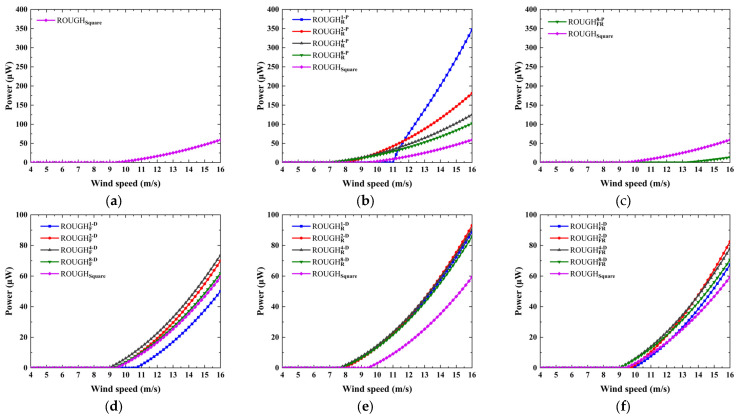
The numerical power versus wind speed: the protruding features on the (**a**) front side only, (**b**) rear side only, and (**c**) both front and rear side; the depressed features on the (**d**) front side only, (**e**) rear side only, and (**f**) both front and rear sides.

## 4. Effects of Aerodynamic Parameters on Performance of GPEH

The aerodynamic coefficient *C_Fy_* is a function of the angle of attack *α*, and therefore of $tan\left(\alpha \right)=\dot{Y}/u$. A polynomial of $\dot{Y}/u$ can be used to approximate this function, as follows:(5)$$C_{Fy}=\sum_{i=0}^{N}A_{i}{\left(\frac{\dot{Y}}{u}\right)}^{i},$$
where the symbol *A_i_* represents the coefficients of the *ith* order ($\dot{Y}/u$) term in the polynomial. Note that if the cross-sectional shape is symmetric in the downstream direction, the polynomial will be an odd function. The *A_i_* of all the considered shapes is obtained using the curve fitting method, as shown in Table A1, Table A2, Table A3, Table A4, Table A5, Table A6, Table A7, Table A8 and Table A9 in Appendix A, and the R-squared is configured to be no less than 0.997 to ensure that the fitted line explains most of the variability of the response data around their means. The parameters used in the model are the same as those of the prototype in Section 2, except for the aerodynamic coefficient. The wind speed step and the maximum calculation time are configured to 0.2 m/s and 100 s, respectively. The convergence condition is that the amplitude error of the displacement response between two adjacent time steps must be less than 10^−5^, or until the calculation time reaches its maximum value. A slight displacement as an initial condition and an updated initial amplitude are also considered.

Critical wind speed and maximum power are usually used as key parameters to evaluate the performance output level of the GPEH, and the meanings of these need to be further expanded. Galloping is a typical aerodynamic instability phenomenon with self-excitation properties, which means that when the wind speed exceeds the critical wind speed, the GPEH produces a relatively high electrical output, and the electrical response level is positively correlated with the galloping amplitude or wind speed. From the perspective of application, the only purpose of structural optimization research on the GPEH is actually to match with the spatiotemporal distribution characteristics of wind energy in a specific geographical area to achieve the best performance. The distribution characteristics can be reflected in parameters such as the average wind speed and the wind speed frequency described by probability density distribution, such as Weibull and Rayleigh distribution. In addition, the influence of material and geometric nonlinearity on the maximum output power under large deformation also needs to be considered. Based on this, in addition to the critical wind speed and maximum output power, the slope of the power versus wind speed curve should also be introduced as one of the performance evaluation parameters. The critical wind speed and maximum output power represent the working wind speed range and the ultimate performance, respectively, and the slope of the power versus wind speed curve represents the rate of the entry into the optimal performance state.

The numerical results of the investigation of power versus wind speed are shown in Figure 9 and Figure 10. The protruding features on the front and rear sides significantly affect the galloping behavior and electrical response of the GPEH, as shown in Figure 9a–c. The introduction of the front protruding features causes the slope of *C_Fy_* versus *α* at *α* = 0° of most of the cross-sectional shapes to be negative. The GPEHs with the above features do not experience galloping and have no electrical output, except for $ROUGH_{FR}^{1-P}$. Compared with $ROUGH_{square}$, $ROUGH_{FR}^{1-P}$ has a higher critical wind speed and lower output power. Therefore, it is necessary to avoid configuring the smooth protruding features on the front side. An interesting performance enhancement comes from the rear protruding features. All four of the considered cases can improve the performance by reducing the critical wind speed and improving the electrical response and the slope of power versus wind speed. The critical wind speed, maximum output power, and the slope *k* of the power versus wind speed curve of $R_{square}$ are 9.40 m/s, 59.29 μW with 16 m/s, and k=1.32u−7.78, respectively. For the cases of $ROUGH_{R}^{1-P}$, $ROUGH_{R}^{2-P}$, $ROUGH_{R}^{4-P}$, and $ROUGH_{R}^{8-P}$, the critical wind speeds are lower than that of the $ROUGH_{square}$ by −17.0%, 14.9%, 25.5%, and 27.6%; the maximum output powers outperform it by 485.9%, 205.4%, 111.2%, 72.4%, and the slopes of the power versus wind speed curve are k=6.50u−22.53, k=3.46u−18.86, k=2.20u−11.45, and k=1.73u−8.64, respectively.

In contrast, the effects of the front and rear depressed features on the galloping behavior and electrical output are relatively weak. All the cases considered produce galloping, and the slope of power versus wind speed is almost consistent with that for the square. Due to the different critical wind speeds, the output power at the same wind speed is different. As shown in Figure 9e, the effect of the depressed feature on the *C_Fy_* versus *α* curve is weak; consequently, the critical wind speed and slope of power versus wind speed are almost the same. The only one advantage is that those cases have a lower critical wind speed than the square, and thus have a relatively higher electrical output.

The effects of the protruding and depressed features on the top and bottom sides on the performance are shown in Figure 10. The protruding features on the top or bottom side only have a significant effect on the critical wind speed and the slope of the power versus wind speed curve, which decreases and increases with the increase in the number of semicircular protrusions, and the performance with those features is inferior to that of the square, as shown in Figure 10a,b. For the protruding features on both the top and the bottom sides, only one case of $ROUGH_{TB}^{1-P}$ can produce galloping oscillation; its critical wind speed is higher and the slope is lower than that of the square. This shows that the configuration of semicircular protruding features on the top or bottom side makes no positive contribution to the improvement of the performance of the GPEH.

Conversely, an important performance improvement comes from the depressed features, as shown in Figure 10c,d. First, the critical wind speed of all the considered cases is lower than that of the square, except for $R_{TB}^{1-D}$, which shows that the introduction of depressed features is an effective way to reduce the critical wind speed. Second, for cases $ROUGH_{T/B}^{1-D}$, $ROUGH_{TB}^{1-D}$, and $ROUGH_{TB}^{2-D}$, a sudden leap of the power versus wind speed curve can be found at the critical wind speed. This is because the obvious depressed features cause a larger peak value and an attack angle corresponding to *C_Fy_* = 0, which, in turn, produces a higher galloping displacement and electrical response amplitude. For example, although the critical wind speed of $ROUGH_{TB}^{1-D}$ is 15.2 m/s, which is higher than the 9.4 m/s obtained for the square, once the wind speed exceeds the critical wind speed, the output power is 288.3 μW (15.4 m/s), which is about 5.59 times that of the square; the critical wind speed of $ROUGH_{TB}^{2-D}$ is approximately 14.9% lower, and the output power at 12 m/s is 68.1 μW and about 310.2% higher than that of the square. Finally, the asymmetric structure in the downstream direction is not considered, primarily, because it has no obvious advantages over the symmetrical structure in terms of critical wind speed and electrical response level.

Based on the aforementioned conclusions, two real-world regions representing wind-rich and wind-poor environments, such as Northern Ireland and Chongqing, China, are, respectively, considered application scenarios for the structural design of the GPEH. The theoretical critical wind speed of galloping is expressed as $u_{g}=\frac{4m\omega \xi }{\rho LDA_{1}}$, where *m* represents the equivalent mass. For low wind speed condition, a GPEH with a lower critical wind speed is essential. Reducing the system’s equivalent mass, damping ratio, and resonant frequency can help achieve this. Consequently, the GPEH’s bluff body and cantilever beam are typically fabricated using lightweight materials and flexible piezoelectric materials, respectively. Additionally, practical constraint by operational space limitation results in a finite system volume. Under identical cross-sectional dimensions, selecting a cross-sectional shape with a higher coefficient *A*_1_ becomes preferable. Among the configurations considered in this study, those with the depressed features on the rear, top, and bottom sides exhibit higher coefficient *A*_1_ compared to the square. The $ROUGH_{TB}^{4-D}$ has the largest coefficient *A*_1_, with a value of 3.77, which is 1.47 times that of the square. Additionally, its slope of power versus wind speed curve is slightly larger than that of the square. For high wind speed condition, rapid transition to optimal operational states is crucial. Due to a sudden leap in the power versus wind speed curve at the critical wind speed, and a steeper slope compared to the square, $ROUGH_{T/B}^{1-D}$, $ROUGH_{TB}^{1-D}$, and $ROUGH_{TB}^{2-D}$ will have greater performance advantages in such application scenario.

## 5. Conclusions

In summary, research on GPEHs with diverse cross-sectional shapes has evolved from elementary shapes (e.g., square, rectangle, triangle) to complex topologies (e.g., D/W/Y/V shapes and circular or square with different rod-shaped attachments). These piecemeal early-stage studies have established a foundational knowledge through a shotgun approach. Although numerous valuable investigations have gradually expanded the understanding of the GPEH, the influence of aerodynamic coefficients primarily determined by cross-sectional shapes on performance still requires systematic and scientific consideration. Investigating methods for configuring cross-sectional shapes or identifying feature variation patterns may be an effective approach, as it helps to obtain details on how continuous feature changes affect performance. This paper provides a preliminary exploration in this research direction.

This work explored the potential effect of the protruding and depressed features on the performance of a galloping piezoelectric energy harvester. It proved that the protruding and depressed features on the bluff body can significantly change the aerodynamic characteristics, galloping behavior, and electrical performance. The evolutionary approach of the bluff body cross-sectional shape was explained, and the aerodynamic lift and drag coefficients of the forty-eight considered cross-sectional shapes were simulated using computational fluid dynamics. An experimentally verified model was used to evaluate the critical wind speed, electrical response level, and the slope of the power versus wind speed curve. There are some remarkable findings in this work, such as the fact that the protruding features should be configured on the rear side of the bluff body, rather than the front, top, and bottom sides. The depressed features should be configured on the top and bottom, rather than the front and rear sides, which is actually an effective method to reduce the critical wind speed. A symmetrical structure of the cross-sectional shape in the downstream direction should be considered as a priority, rather than an asymmetric structure. Compared with the square cross-sectional shape, there are two unconventional cases in terms of output power among all of those considered, $ROUGH_{R}^{1-P}$ and $ROUGH_{TB}^{1-D}$, and a significantly larger slope of the power versus wind speed curve was found for those two cases, respectively; thus, they represent a significant positive contribution to the optimization of high-performance harvester designs. In the future, non-semicircular protruding and depressed features, such as triangular shapes, and the height or depth of protruding and depressed features need to be considered to enhance the performance of a galloping-based piezoelectric energy harvester.

## Figures and Tables

**Figure 1 micromachines-16-00254-f001:**
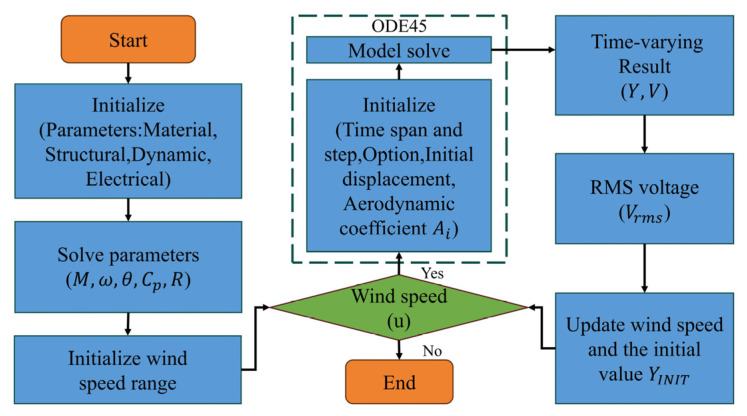
The flow chart of the simulation process.

**Figure 2 micromachines-16-00254-f002:**
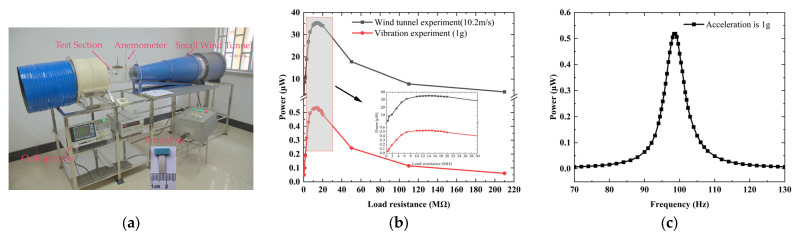
(**a**) The fabricated prototype; (**b**) experimental power versus load; (**c**) experimental power versus frequency.

**Figure 3 micromachines-16-00254-f003:**
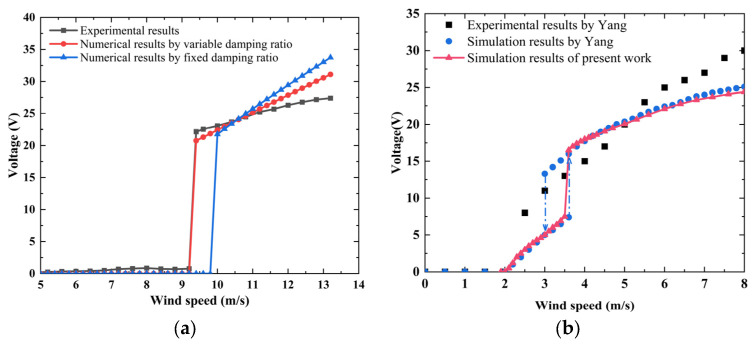
The experimental and numerical results: (**a**) prototype in this work; (**b**) previous study by Yang [22].

**Figure 4 micromachines-16-00254-f004:**
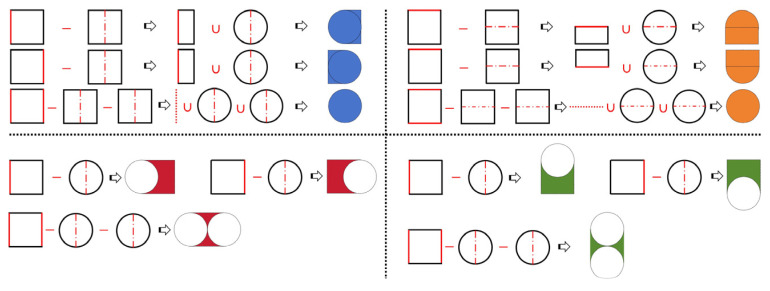
Venn diagram of the bluff body cross-sectional shape evolutionary approach.

**Figure 5 micromachines-16-00254-f005:**
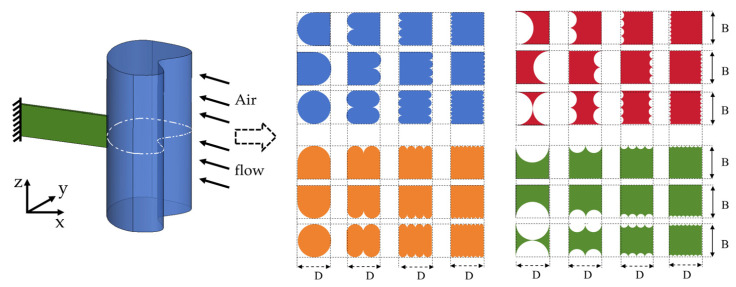
The cross-sectional shapes of the bluff bodies used in this present work.

**Figure 6 micromachines-16-00254-f006:**
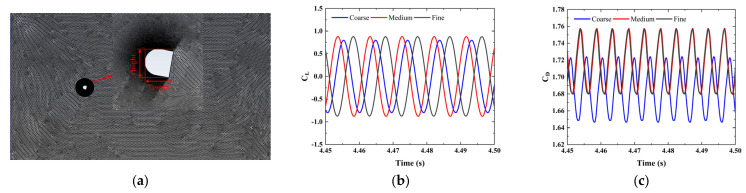
(**a**) Computational mesh, comparison of results in time domain: (**b**) *C_L_*, (**c**) *C_D_*.

**Figure 10 micromachines-16-00254-f010:**
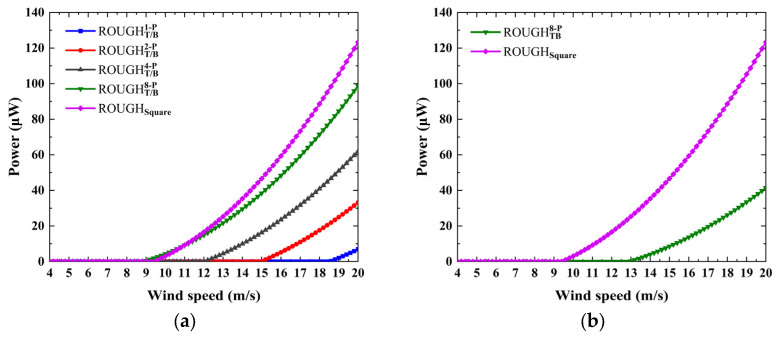
The numerical power versus wind speed: the protruding features on the (**a**) top or bottom side only, (**b**) both top and bottom sides; the depressed features on the (**c**) top or bottom side only, (**d**) both top and bottom sides.

**Table 1 micromachines-16-00254-t001:** Comparison of results of the aerodynamic lift and drag coefficients.

Case	Grid Number	*C_L_* _(*RMS*)_	*C_D_* _(*RMS*)_
Coarse	$3.72\times 10^{4}$	0.566	1.683
Medium	$1.03\times 10^{5}$	0.622 [9.89%]	1.716 [1.90%]
Fine	$2.34\times 10^{5}$	0.624 [0.32%]	1.716 [0.00%]

## Data Availability

Data are contained within the article.

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
