# Peer review of "Effects of Aerodynamic Parameters on Performance of Galloping Piezoelectric Energy Harvester Based on Cross-Sectional Shape Evolutionary Approach"

_micromachines, 2025, doi:10.3390/mi16030254_

Round 1

Reviewer 1 Report

Comments and Suggestions for Authors This paper investigates the effects of aerodynamic parameters on the performance of galloping-based piezoelectric energy harvesters (GPEH). It proposes a cross-sectional shape evolution method based on Boolean operations, which generates 48 different cross-sectional shapes by modifying protruding and depressed features on a square cross-section. This method systematically explores the impact of aerodynamic parameters on GPEH performance. The accuracy of the theoretical model is verified through wind tunnel experiments, with results showing good agreement between experimental and numerical simulation outcomes, further demonstrating the reliability of the theoretical model. The paper is innovative, and specific suggestions for improvement are as follows:
  1. The study currently focuses on protruding and depressed features in the shape of semicircles. It is recommended to consider other shapes (such as triangles and rectangles) in future research to further explore their impact on performance.
  2. The paper mainly focuses on performance optimization under laboratory conditions. It is suggested to add an analysis of adaptability to real-world application scenarios (such as outdoor environmental monitoring) in the discussion section, considering factors like wind speed variations and environmental disturbances that may affect performance.
  3. Although the paper mentions that material and geometric nonlinearities have some impact on performance, these factors are not thoroughly analyzed. It is recommended to conduct more detailed modeling and experimental verification of nonlinear factors in future research to improve the theoretical framework.
  4. Some of the figures in the paper (such as Figures 7 and 8) are relatively complex. It is suggested to optimize the design of these figures, add legends and annotations, to help readers better understand the data and conclusions.
  5. The paper could benefit from a comparative analysis with other similar studies in the discussion section to highlight the innovations and advantages of this research, as well as to point out directions for future work.

Author Response

请参阅附件。

Reviewer 2 Report

Comments and Suggestions for Authors

The article investigates the influence of aerodynamic parameters of a bluff body’s cross-sectional shape on the performance of a galloping piezoelectric energy harvester (GPEH). Overall, the study is well-structured, and the computational scheme, along with the main concept of the evolutionary algorithm, is presented clearly. The numerical model used for GPEH simulations has been validated experimentally, and an appropriate convergence analysis was conducted for the aerodynamic model in the finite element calculations. The introduction provides sufficient background information to understand the fundamental aspects of the problem and the current state of research in the field. The references primarily include recent works, contributing to the relevance of the study.

Some things are not completely clear.

What justifies the high load resistance values at which the maximum power output is achieved?

It remains unclear whether the proposed evolutionary approach was implemented as an algorithm or if the optimization process was conducted through exhaustive enumeration of all possible shapes.

Does the methodology involve solving the aerodynamic problem in Ansys to determine the A_i coefficients for each cross-sectional shape, which were subsequently used in the electroelastic analysis? If so, was this process automated in any way?

Reviewer 3 Report

Comments and Suggestions for Authors

The present manuscript carries the following title: "Effects of Aerodynamic Parameters on Performance of Galloping Piezoelectric Energy Harvester Based on Cross-sectional Shape Evolutionary Approach". 

It presents a study on harvesting vibrational energy due to galloping using a PVDF film.

Most of the paper deals with aerodynamic investigation of the body used to produce galloping movement aimed at exiting the harvester.

In general, the article is well written, with two minor issues to be clarified:

  1. In view of a previous study of the authors, see Ref. 35, what is new in the present manuscript and what has been taken from Ref. 35 ? Please comment on this issue.
  2. The power harvested is in the range of micro watts, as shown in Fig.  2c, so the authors are asked to provide some reasons for using this type of harvester which yields a very low power. This is important to assess the feasibility of such harvester.
